

# Surfactants regulate the mixing state of organic-inorganic mixed aerosols undergoing liquid-liquid phase separation

Younuo Fan,[a] Qiong Li,[a,b,*] Min Zhou,[a,c] Jiuyi Sun,[d] Shuaishuai Ma,[a,*] Tianyou Xu[a]

[a] *College of Chemical and Material Engineering, Quzhou University, Quzhou 324000, China*

[b] *Shanghai Key Laboratory of Atmospheric Particle Pollution and Prevention, Department of Environmental Science & Engineering, Fudan University, Shanghai 200433, China*

[c] *College of Chemical Engineering, Zhejiang University of Technology, Hangzhou 310000, China*

[d] *The Institute of Chemical Physics, School of Chemistry and Chemical Engineering, Beijing Institute of Technology, Beijing 100081, China*

*Correspondence to*: Qiong Li (lq270224181@126.com) and Shuaishuai Ma (mass@qzc.edu.cn)

**Abstract.** The mixing state of atmospheric aerosols undergoing liquid-liquid phase separation (LLPS) is crucial for regulating atmospheric chemistry and influencing global climate, often adding uncertainties to atmospheric and climate models. Despite its significance, understanding how coexisting species, such as surfactants, affect the mixing state of phase-separated aerosols remains limited. This study investigated the phase transition behaviors and resulting mixing states of

aerosols composed of 1,2,6-hexanetriol and ammonium sulfate, with added surfactants. Contrary to the commonly assumed core-shell structure, we observed that at very low concentrations of hydrocarbon surfactants, the organic phase partially engulfed the aqueous inorganic phase, a configuration we termed partial organic-phase engulfing. Furthermore, we discovered the formation of partial inorganic-phase engulfing and inverse core-shell structures, where the inorganic phase partially or fully spread over the organic domain at higher surfactant levels. We identified a relationship between equilibrium

particle morphology and spreading coefficients, primarily governed by surface tension reduction in the separated organic and inorganic phases. These distinctive mixing states may substantially alter the chemical, physical, and optical properties of organic-inorganic aerosols under real atmospheric conditions. Our findings bridge a critical knowledge gap regarding the role of surface tension evolution in the equilibrium particle morphology of internally mixed atmospheric particles and its potential impact on aerosol-chemistry-climate interactions. These insights emphasize the need to refine current aerosol

models to incorporate the specific LLPS morphologies observed in this study.



## 1 Introduction

Atmospheric aerosols consist of a diverse array of organic and inorganic species (Murphy et al., 2006; Lee et al., 2002; Huang et al., 2014). These mixed organic-inorganic aerosols can undergo phase transitions, such as liquid-liquid phase separation (LLPS), efflorescence during dehumidification, or deliquescence upon humidification. The resulting mixing states, such as liquid-liquid or liquid-solid equilibria, significantly influence aerosol surface properties, water uptake kinetics (Davies et al., 2013; Mikhailov et al., 2021), gas-particle partitioning of semi-volatile organic compounds (SVOCs) (Ohno et al., 2022; Schervish and Shiraiwa, 2023; Shiraiwa et al., 2013; Zuend and Seinfeld, 2012; Choczynski et al., 2024), heterogeneous chemical reactions (Cosman et al., 2008; Shen et al., 2022; Drozd et al., 2013), and solar radiative forcing (Yu et al., 2005; Martin et al., 2004; Fard et al., 2018; Zhang et al., 2022).

The salting-out effect induces non-ideal thermodynamic behavior in the organic and inorganic components of internally mixed aerosols at the phase separation relative humidity (SRH), leading to the formation of a strongly polar inorganic-rich phase and a weakly polar organic-rich phase, a process known as LLPS (Marcolli and Krieger, 2006). A commonly observed morphology following LLPS is the core-shell structure, where the aqueous electrolyte phase is fully enclosed by the outer organic shell. The oxygen-to-carbon elemental ratio (O:C) of organic components indicates the polarity of organic molecules and their miscibility with inorganic salts and water, thereby serving as a predictor for LLPS and the hygroscopic properties of mixed systems (Song et al., 2012b; Bertram et al., 2011; Malek et al., 2023). Notably, the aromaticity of organic species has also been identified as a key regulator of LLPS, with increased aromatic oxidation generally promoting phase separation (Tackman et al., 2024). Furthermore, the predictive capability of the O:C ratio becomes limited in systems containing nitrogen-containing organics, for which the X:C ratio (i.e., both O:C and N:C) may serve as a more suitable proxy (Ferdousi-Rokib et al., 2025). In addition, several other factors such as the organic fraction (Ciobanu et al., 2009; Bertram et al., 2011; Song et al., 2012a), inorganic species (You et al., 2013), drying rate (Altaf and Freedman, 2017), particle size (Ohno et al., 2023; Freedman, 2020; Kucinski et al., 2019), and bulk viscosity (Ye et al., 2025), can also influence LLPS occurrence.

However, limited studies have focused on the mixing states of biphasic organic-inorganic particles. Reid's group proposed that water-insoluble organics with low O:C ratios, when mixed with aqueous inorganics, may form a partial engulfing structure in which the aqueous inorganic phase is partially enclosed by an organic lens; meanwhile, high surfactant concentrations further inhibit the spreading of organics on aqueous droplets (Reid et al., 2011; Kwamena et al., 2010). Qiu and Molinero (2015) developed a predictive model for the equilibrium morphology of liquid-liquid aerosols, quantifying the contact angle between phases based on Young's equation and free energy minimization. In contrast, internally mixed organic-inorganic systems, where oxidized secondary organic aerosol (SOA) serves as the organic phase, predominantly adopt a core-shell morphology (Gorkowski et al., 2020; Song et al., 2013; Ciobanu et al., 2009; Bertram et al., 2011). Only several studies have reported partial engulfing structures in such systems, and the underlying mechanisms remain unclear (Song et al., 2012a; Veghte et al., 2013; Kucinski et al., 2020).





Surface-active organic matter, derived from both anthropogenic and biogenic sources (Franklin et al., 2022; Latif and
Brimblecombe, 2004), is present in sea surface organic films and can be transported into the atmosphere via ocean bubble
bursting (Liss and Duce, 1997; Van Acker et al., 2021; Liu et al., 2022; Kroflič et al., 2018), leading to a 15-25-fold
enrichment of surfactants in sea spray aerosol (SSA) (Jayarathne et al., 2016; Russell et al., 2010; Donaldson and Vaida,
2006). $PM_{2.5}$ samples collected at the Baltic station of Askö, Sweden, exhibited high surfactant concentrations of at least 27
± 6 mM, with surface tension values below 40 mN·m$^{-1}$ (Gérard et al., 2016). Smaller atmospheric particles generally contain
higher concentrations of surface-active organics (Kroflič et al., 2018). Biogenic organic aerosols, such as pollen (Prisle et al.,
2019), common urban aerosol organics like dicarboxylic acids and humic acid (Taraniuk et al., 2007; Fan et al., 2024), and
per- and polyfluoroalkyl substances (PFAS) (Barbosa et al., 2024), frequently detected in the sea surface microlayer (Casas
et al., 2020), airborne particles (Li et al., 2024b), and indoor dust (Tan et al., 2024), also demonstrate surface activity.
Surfactants tend to form a thin organic layer in an "inverted micelle" configuration on aerosol surfaces, affecting molecular
transport across the aqueous-air interface and particle surface reactions (Faust and Abbatt, 2019; Miñambres et al., 2014;
Xiong et al., 2023; Harmon et al., 2010). For example, sodium dodecyl sulfate (SDS) can inhibit the hygroscopic growth of
NaCl nanoparticles due to its surface monolayer (Harmon et al., 2010), or suppress the photochemical aging of propionic
acid droplets (oxidation products of α-pinene) by approximately 60% in reactions with OH radicals (Faust and Abbatt, 2019).
Surface-active organics can also reduce droplet surface tension, lowering the critical supersaturation required for particle
activation into cloud droplets (Lowe et al., 2019; Liu et al., 2018; Hartery et al., 2022; Prisle, 2021; Ovadnevaite et al., 2017).
Field and model studies indicated that reduced surface tension due to LLPS and the resulting organic surfactant shell
significantly increased cloud condensation nucleus (CCN) concentrations, more than compensating for the Raoult effect
(Ovadnevaite et al., 2017). This in turn highlights the critical role of morphology evolution in altering aerosol surface tension
and CCN activity.

In internally mixed atmospheric aerosols, surfactants are expected to reduce the surface and interfacial tensions of the
organic and inorganic phases, altering their spreading coefficients and affecting the mixing state of biphasic systems.
Furthermore, the salting-out effect can enhance surfactant partitioning to the surface, with bulk-phase depletion further
perturbing surface tension in salt-containing aqueous droplets (Bzdek et al., 2020; Prisle et al., 2019). These scenarios
suggest more complex solute-solute and solute-water interactions, as well as non-ideal thermodynamic behavior during
LLPS. Therefore, comprehensive laboratory studies are urgently needed to clarify how surfactants influence LLPS processes
and the equilibrium morphology of internally mixed organic-inorganic particles. 1,2,6-hexanetriol, a polyhydroxy organic
compound with an O:C ratio of 0.5, serves as a chemical mimic of typical SVOCs in the atmosphere (Lv et al., 2019) and
can represent moderately oxidized SOA (Canagaratna et al., 2015). Our previous work employed the 1,2,6-hexanetriol and
ammonium sulfate (AS) model system to investigate the dynamic process and mechanism of LLPS (Ma et al., 2021). In the
present study, SDS was selected as a surrogate for atmospheric soluble surfactants due to its structural similarity to naturally
occurring long-chain monocarboxylic acids, such as lauric acid ($C_{12}$) (Mayol-Bracero et al., 2001; Gill et al., 1983), and its
well-established surface properties comparable to those of natural surfactants (Gérard et al., 2016; Petters and Petters, 2016).



We explored the phase transition behaviors, particularly LLPS, and the equilibrium particle morphology of 1,2,6-hexanetriol/AS mixed particles in the presence of SDS using microscopic imaging and confocal Raman spectroscopy.
Furthermore, we systematically analyzed the influence of the organic-to-inorganic molar ratio (OIR), surfactant type, and the associated mechanisms, including the role of spreading coefficients.

## 2 Experimental Section

### 2.1 Sample preparation

All chemicals were obtained from Shanghai Macklin Biochemical Technology Co., Ltd., including AS ($\geq$ 99%), 1,2,6-
hexanetriol ($\geq$ 97%), SDS (92.5-100.5%), cetyltrimethyl ammonium chloride (CTAC, $\geq$ 99%), Triton X-100 ($\geq$ 98%), sodium dodecylbenzene sulfonate (SDBS, $\geq$ 95%), dioctyl sodium sulfosuccinate (AOT, $\geq$ 96%), benzethonium chloride (Hyamine, $\geq$ 97%), and Tergitol NP-40. Surfactant solutions with varying initial concentrations were prepared by dissolving the reagents in ultrapure water (18.2 M$\Omega$ cm resistivity). 1,2,6-hexanetriol and AS at aqueous concentrations of 0.1 or 0.4 mol L$^{-1}$, corresponding to OIRs of 1:4, 1:1, and 4:1, were dissolved in the surfactant solutions. Aqueous droplets with
specific molar ratios of 1,2,6-hexanetriol/AS/surfactant were then sprayed onto a hydrophobic fluorinated ethylene propylene (FEP) film using a microsyringe.

### 2.2 Microscopic imaging observations of single particles

The microscopic imaging method have been described in detail previously (Ma et al., 2021; Wang et al., 2022); thus, only a brief overview is provided here. An inverted optical microscope (Aosvi BX19-HK830, 25× objective, 0.40 numerical
aperture) coupled with a CCD camera was used to observe the hygroscopic growth and phase transitions of mixed aerosols deposited on the FEP substrate, which was fixed to the bottom glass slice of the sample cell (~20 cm$^3$). The sample cell was sealed with a transparent glass slice on top. The ambient relative humidity (RH) around the deposited particles was adjusted using dry and wet N$_2$ streams at varying flow rate ratios, monitored in real-time with a hygrometer (Centertek Center 313, accuracy ± 2.5%). The RH was continuously varied at an average rate of 0.06-0.07% RH s$^{-1}$ within a range of approximately
5-95%. The sizes of deposited particles at 90% RH were in the vicinity of 100 μm. The geometric areas of the recorded images of deposited particles under different RH values were determined using image analysis software (ToupView X64). The area ratio during an RH cycle was calculated as follows:

$$Area\ ratio = \frac{A_{RH}}{A_0} \tag{1}$$

where $A_{RH}$ represents the geometric area of the particles at a given RH, while $A_0$ denotes the geometric area of effloresced
particles at RH < 10%.





### 2.3 *In situ* Raman measurements of single particles

Raman spectra of SDS-doped particles were acquired using a Renishaw InVia confocal Raman spectrometer coupled with a Leica DMLM microscope (50× objective, 0.75 numerical aperture) (Wang et al., 2017; Zhou et al., 2014; Wang et al., 2005).

An argon-ion laser with a wavelength of 514.5 nm served as the excitation source, operating at 20 mW with a diffraction grating density of 1800 grooves per millimeter. Aqueous droplets deposited on the FEP film were positioned at the bottom of the sample cell. The ambient RH was adjusted and monitored similarly to the microscopic observations. Raman spectra were collected from different locations on the droplets at specific RH values to obtain spatially resolved component distributions. The spectra ranged from 800 to 4000 cm$^{-1}$ with a resolution of 1 cm$^{-1}$ and were obtained through two scans, each with a 10 s

acquisition time. All measurements were conducted at room temperature (23–26 °C).



# 3 Results and Discussion

## 3.1 Phase transition behaviors of OIR = 1:1 particles at varying initial SDS concentrations

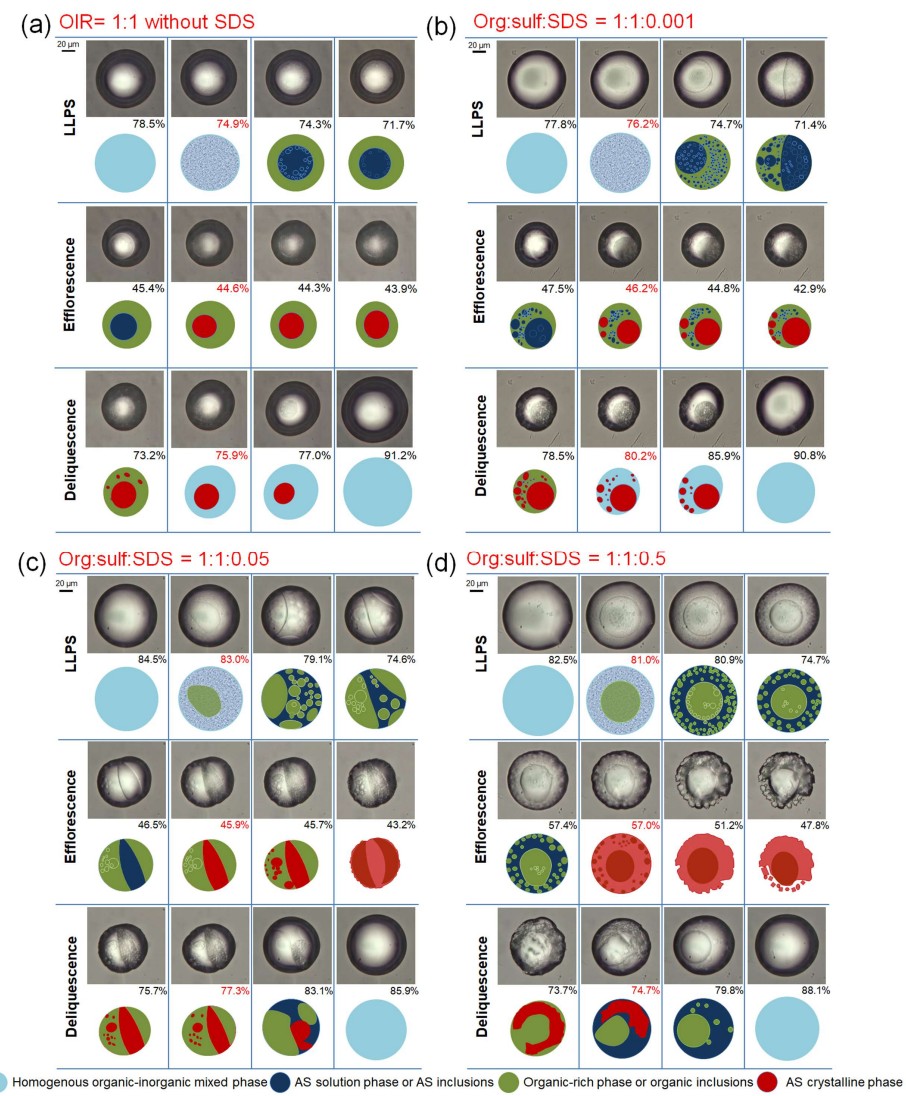





**Figure 1.** Optical images and corresponding illustrations of mixed 1,2,6-hexanetriol/AS particles with an OIR of 1:1, without SDS (a), and

with org:sulf:SDS ratios of 1:1:0.001 (b), 1:1:0.05 (c), and 1:1:0.5 (d), during LLPS, efflorescence, and deliquescence. The RH percentage

is indicated in each frame, with specific values highlighted in red to signify the occurrence of phase transitions.

Figure 1 compares the phase transition behaviors, including LLPS, efflorescence, and deliquescence, of a mixed 1,2,6-hexanetriol/AS particle without SDS and that with a org:sulf:SDS molar ratio of 1:1:0.001. The corresponding hygroscopic growth curves are shown in Figure S1. For the SDS-free OIR = 1:1 particle, schlieren patterns suddenly appear at the SRH of

74.9%, followed by the growth and coalescence of dispersed clusters, ultimately forming a core-shell morphology with an inner AS solution core and an outer organic-rich shell (Figure 1a). This spinodal decomposition mechanism aligns with our previous observations (Ma et al., 2021). As the RH decreases, the AS solution phase crystallizes at the efflorescence relative humidity (ERH) of 44.6%, causing a significant area ratio reduction (Figure S1a). During humidification, the AS crystal begins to dissolve at the deliquescence relative humidity (DRH) of 75.9%, evidenced by a sharp increase in area ratio above

this RH. The outer aqueous 1,2,6-hexanetriol phase continuously absorbs and releases water without undergoing phase transitions (Ma et al., 2021).

     For the org:sulf:SDS = 1:1:0.001 particle, LLPS occurs at 76.2% RH via spinodal decomposition (Figure 1b). However, instead of the expected fully engulfed core-shell structure, a partial engulfing morphology is observed at 71.4% RH. Here, an organic phase lens partially encloses the inorganic AS solution, which remains in contact with the gas phase, a configuration

referred to as partial organic-phase engulfing. This indicates that the core-shell structure may not always be the thermodynamically favored state for internally mixed aerosols undergoing LLPS, particularly in the presence of surfactants. The distinction between phases is primarily determined by efflorescence behavior here, i.e., the engulfed AS solution phase crystallizes at 46.2% RH, while the surrounding organic phase remains liquid during drying. The SDS molar fraction of 0.05% (org:sulf:SDS = 1:1:0.001) marks the threshold concentration determining equilibrium morphology, transitioning from core-

shell to partial engulfing. As shown in Figure S2, mixtures with lower SDS fractions (1:1:0.0005 and 1:1:0.00075) retain a core-shell structure after LLPS, whereas higher fractions induce partial engulfing. Moreover, the SDS concentration threshold varies with OIR, as seen in OIR = 1:4 and 4:1 particles (Section S1). Both LLPS mechanisms, i.e., growth of a second phase from the particle surface for OIR = 1:4 and nucleation and growth for OIR = 4:1 (Ma et al., 2021; Ciobanu et al., 2009), shift toward spinodal decomposition in the presence of SDS.

As the SDS fraction increases, the partial engulfing structure is maintained in the 1:1:0.005 and 1:1:0.01 mixtures (Figure S7). Notably, in the 1:1:0.01 mixture, the area of the inorganic phase exposed to the gas phase increases, resulting in irregular particle morphology after efflorescence. When the SDS molar fraction reaches 2.38% (org:sulf:SDS = 1:1:0.05), the aqueous inorganic phase becomes continuous, and the organic inclusions form a dispersed phase during LLPS (Figure 1c). The phase-separated particle transitions to a state of partial inorganic-phase engulfing, where the organic phase is

partially surrounded by the continuous inorganic phase. In a core-shell structure, the organic phase, characterized by lower surface tension, typically occupies the outer shell to minimize total free energy. However, as the SDS concentration increases, the reduced inorganic-air surface tension increases the surface area of the inorganic phase exposed to the gas phase,





minimizing the total Gibbs energy of the mixed systems (Reid et al., 2011). Consequently, further increasing the SDS concentration (as in the 1:1:0.5 mixture) results in complete engulfing of the organic phase by the outer inorganic phase after
LLPS, forming an inverse core-shell structure (Figure 1d). Additionally, the LLPS morphologies of the 1:1:0.1 and 1:1:1 mixtures are identified as partial inorganic-phase engulfing and inverse core-shell, respectively (Figure S8).

To further evaluate the relevance of these findings to atmospheric micron-sized aerosols, we investigated LLPS morphologies of 1,2,6-hexanetriol/AS/SDS particles with smaller sizes (~8–20 μm). As shown in Figure S9, the observed morphologies include core-shell (1:1:0.001), partial organic-phase engulfing (1:1:0.005), partial inorganic-phase engulfing
(1:1:0.2), and inverse core-shell (1:1:0.5). However, achieving these morphologies in smaller particles requires higher SDS fractions than in supermicrometer particles, likely due to weaker solute effects and increased surface tensions resulting from their higher surface-to-volume ratios (Bzdek et al., 2020). Given the acidic nature of marine and urban aerosols (Pye et al., 2020; Angle et al., 2021; Zheng et al., 2020), we further examined phase transitions of 1,2,6-hexanetriol/AS/SDS particles derived from mixture solutions at pH 1.5 (Figure S10). Notably, even a low SDS molar fraction of 0.50% can induce an
inverse core-shell structure in the 1:1:0.01 mixture, implying that this previously overlooked morphology may be common in surfactant-doped acidic aerosols. Moreover, the prevalence of partial engulfing structures was also verified in other organic/AS/SDS mixed systems, such as those with diethyl decanedioate, 1,2-hexanediol, and 3,3-dimethylglutaric acid, which exhibit O:C ratios of 0.29, 0.33, and 0.57, respectively (see Figure S11). However, the detailed effects of aerosol pH, particle size, and diverse organic and inorganic compositions fall beyond the scope of this study and will be expatiated in
future work.




### 3.2 Spatially resolved Raman analysis of SDS-doped particles

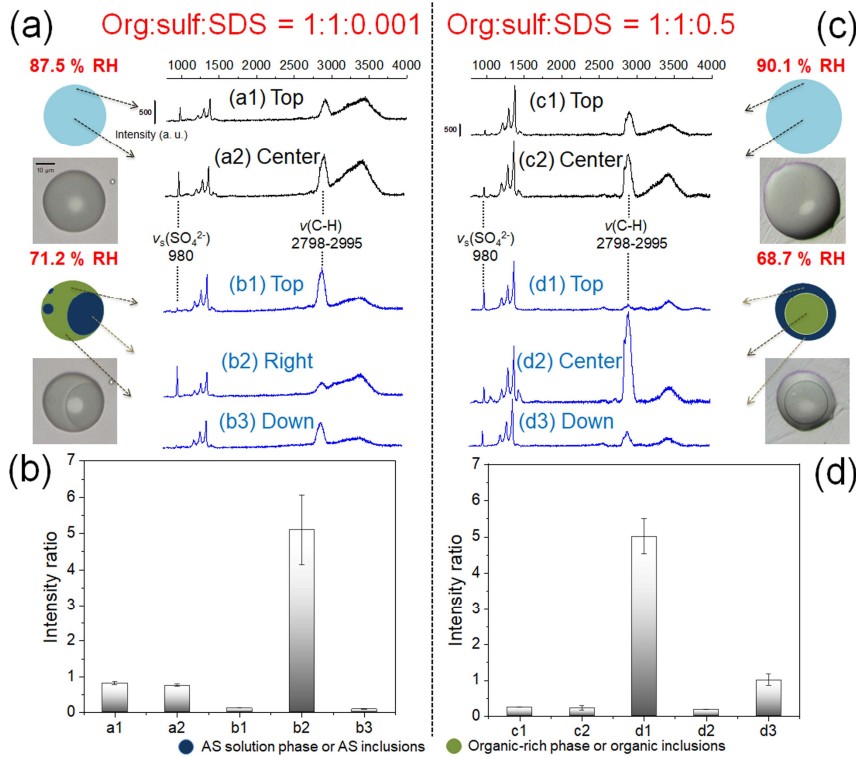

**Figure 2.** Spatially resolved Raman measurements of different locations on the org:sulf:SDS = 1:1:0.001 (a) and 1:1:0.5 (c) particles during dehumidification. The intensity ratios of the $\nu_s(SO_4^{2-})$ band to the $\nu$(C-H) band for the 1:1:0.001 (b) and 1:1:0.5 (d) particles are also

presented. Error bars in panels (b) and (d) were obtained based on triplicate measurements. The corresponding optical images and illustrations are shown alongside the Raman spectra. Note that the labeled spatial orientation corresponds to the direction observed from the microscopic perspective.

To confirm the partial engulfing and inverse core-shell structures of SDS-doped particles, spatially resolved Raman spectra were collected for org:sulf:SDS = 1:1:0.001 and 1:1:0.5 particles under specific RH conditions, as depicted in Figure 2a and

2c. The spatial distribution of organic and inorganic components was determined using the intensity ratio of feature bands for AS and 1,2,6-hexanetriol (Figure 2b and 2d). The symmetric stretching vibration of $SO_4^{2-}$ ($\nu_s(SO_4^{2-})$) appears at 980 cm$^{-1}$, while the C-H stretching vibration ($\nu$(C-H)) of 1,2,6-hexanetriol is observed between 2798 and 2995 cm$^{-1}$. At 87.5% RH, the 1:1:0.001 particle exhibits a homogeneous mixed phase, as indicated by the similar intensity ratios of spectra a1 and a2 (0.81



and 0.75, respectively). As RH decreases to 71.2%, a partial engulfing structure emerges. Spectra b1 and b3, from the
organic-rich phase, show lower intensity ratios of 0.11 and 0.09, whereas spectrum b2, corresponding to the engulfed
inorganic phase, exhibits a significantly increased intensity ratio of 5.09. This confirms the distinct separation of organic and
inorganic components. However, small amounts of AS and 1,2,6-hexanetriol remain in the organic and inorganic phases,
respectively, as evidenced by weak signatures of their feature bands. The O-H stretching vibration ($\nu$(O-H)) around 3400 cm$^{-1}$ in spectra b1 and b3 indicates the presence of liquid water in the organic-rich phase.

For the 1:1:0.5 particle, spectra c1 and c2 display nearly identical intensity ratios (0.24 and 0.22). After LLPS, the
intensity ratios of spectra d1 and d3, obtained from the top and down of the particle, increase significantly to 5.01 and 1.00,
respectively, while the intensity ratio of d2, acquired from the center, decreases to 0.18, confirming the formation of an
inverse core-shell morphology. Indeed, the high SDS content in the 1:1:0.5 mixture may have influenced the peak intensities
of both $\nu_s(SO_4^{2-})$ and $\nu$(C-H). However, the slight deviations did not affect the overall interpretation of particle morphology,
which was also supported by optical images and efflorescence behavior (see Figure 1d).





**3.3 Surface tension and spreading coefficients for phase-separated biphasic particles**

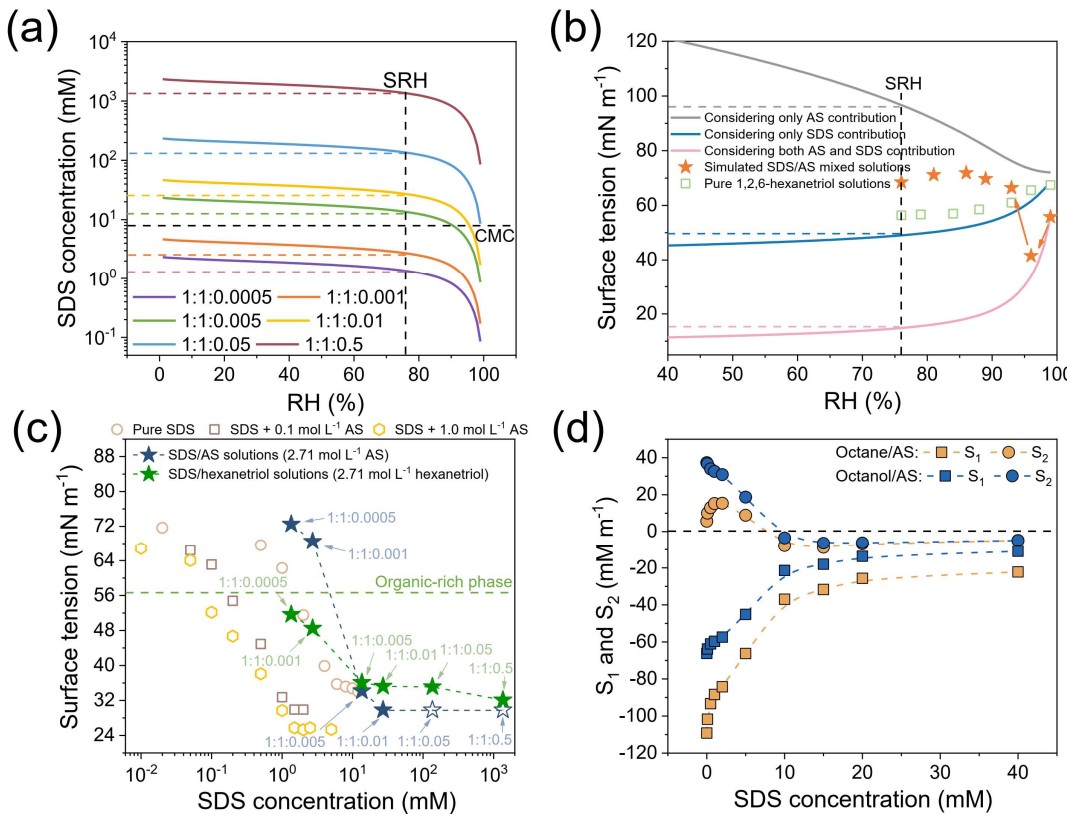

**Figure 3.** (a) Evolution of SDS concentrations with decreasing RH, as predicted by the EAIM model for OIR = 1:1 particles with varying initial SDS concentrations. (b) Surface tension of org:sulf:SDS = 1:1:0.001 aqueous droplets as RH decreases, with predictions considering only AS, only SDS, or both AS and SDS contributions shown as lines and measurements from simulated SDS/AS mixed solutions and pure 1,2,6-hexanetriol solutions represented as scatter points. (c) Surface tension measurements at varying SDS concentrations for SDS/AS and SDS/1,2,6-hexanetriol mixed solutions, with a fixed AS or 1,2,6-hexanetriol concentration of 2.71 mol L$^{-1}$. Hollow pentacles indicate estimated approximate surface tension values, and the green dotted line corresponds to the surface tension of aqueous 1,2,6-hexanetriol solution at 2.71 mol L$^{-1}$. (d) Calculated spreading coefficients ($S_1$ and $S_2$) as a function of SDS concentration for octane/AS and octanol/AS model systems.

The relative molar ratio of organic or inorganic solutes to surfactant in macroscopic solutions was assumed to be conserved in deposited droplets during RH fluctuations. Using the Extended Aerosol Inorganic Model (EAIM), we quantified the SDS concentration increases with decreasing RH in OIR = 1:1 particles with varying initial SDS levels (Figure 3a), which does



not account for efflorescence behavior. This estimation was based on the Universal Quasi-Chemical Functional Group

Activity Coefficient (UNIFAC) method, in which the functional groups of 1,2,6-hexanetriol were specified as 1 CH, 5 CH₂, and 3 OH. Note that water uptake by the less hygroscopic SDS was negligible; therefore, according to the Zdanovskii-Stokes-Robinson (ZSR) relation (Stokes and Robinson, 1966; Malm and Kreidenweis, 1997), the solute concentrations in ternary mixed droplets were assumed to be unaffected by the SDS component. The SRH for different mixed systems was approximated to be 76%, corresponding to an AS and 1,2,6-hexanetriol concentration of 2.71 mol L$^{-1}$ at this SRH for these

OIR = 1:1 particles based on EAIM predictions. In the 1:1:0.001 mixture, a low SDS concentration of 2.71 mM, below the critical micelle concentration (CMC) of pure SDS solutions (~8 mM) (Wangsakan et al., 2001; Rana et al., 2002), significantly influenced equilibrium particle morphology.

Figure 3b presents the surface tension simulations for org:sulf:SDS = 1:1:0.001 droplets as RH decreases, comparing predictions based on AS alone, SDS alone, and both components, alongside measurements from simulated SDS/AS mixed

solutions and pure 1,2,6-hexanetriol solutions. Details of the surface tension predictions are provided in Section S2. When considering only AS, surface tension increases significantly from 72.04 to 96.68 mN m$^{-1}$ as RH decreases from 99% to 76%, aligning with previous studies (Boyer et al., 2017). Predictions based solely on SDS show a surface tension reduction to 49.06 mN m$^{-1}$ at 76% SRH, following the empirical Szyskowski equation. Incorporating both AS and SDS contributions using a modified Szyskowski equation reveals that inorganic solutes enhances bulk-to-surface partitioning of SDS due to the

salting-out effect, further reducing surface tension to 14.84 mN m$^{-1}$ at the SRH. Measurements for 1,2,6-hexanetriol systems, obtained via the hanging droplet method (KRÜSS DSA 25S, Germany), show an ~11 mN m$^{-1}$ decrease from 99% to 76% RH. To simulate the surface tension evolution of org:sulf:SDS = 1:1:0.001 droplets, we used SDS/AS mixed solutions corresponding to EAIM-predicted solute concentrations. Note that the surface tension of finite-sized droplets, especially those with radii < 10 μm, is size-dependent and typically higher than that of macroscopic solutions. This arises from the

bulk-to-surface partitioning of surfactants, driven by the high surface-to-volume ratios of aqueous droplets (Bzdek et al., 2020; Prisle et al., 2019; Bain et al., 2023). However, given the larger particle sizes and higher solute concentrations studied here, deviations in surfactant bulk concentration and surface tension between supermicrometer droplets and macroscopic solutions are expected to be minor (Bianco and Marmur, 1992; Laaksonen, 1993; Malila and Prisle, 2018). Nevertheless, the solution-phase surface tension values measured here should be regarded only as approximate estimates, intended to relate

LLPS morphology to spreading coefficients, and likely deviate from the actual surface tension of smaller atmospheric aerosol droplets. As shown in Figure 3b, the simulated surface tension for mixed solutions decreases to 41.55 mN m$^{-1}$ at 96% RH, following the modified Szyskowski trend. As RH decreases further to 93%, the surface tension rises to 66.43 mN m$^{-1}$, deviating from predictions. This is because the Szyskowski equation predicts surface tension accurately in dilute solutions, but it fails at higher or supersaturated solute concentrations. After minor fluctuations, surface tension stabilizes at 68.49 mN

m$^{-1}$ at the SRH. These deviations can be attributed to the competing effects of concentrated inorganic ions and surfactant molecules on surface tension. Increasing AS concentration tends to elevate surface tension due to attractive ion-water interactions (Boyer et al., 2017), whereas higher SDS concentrations and the salting-out effect lower it.





The spreading coefficients, $S_i$, for phases $i$ and $j$ within a third phase $k$ predict mixing states in phase-separated multiphase systems and are expressed as follows (Reid et al., 2011; Kwamena et al., 2010; Li et al., 2022; Qiu and Molinero, 2015; Gorkowski et al., 2020):

$$S_i = \sigma_{jk} - (\sigma_{ij} + \sigma_{ik}) \qquad (2)$$

Here, the inorganic, organic, and gas phases are labeled as phases 1, 2, and 3, respectively. Therefore, $S_1$ reflects the spreading of the inorganic phase on the organic liquid, while $S_2$ represents the spreading of the organic phase on the inorganic phase. For two immiscible phases, the energy recovered from the loss of the organic-air surface is typically insufficient to offset the cost of forming a larger inorganic/organic interface and an inorganic surface, resulting in generally negative $S_1$ values (Reid et al., 2011; Kwamena et al., 2010). Conversely, the energy recouped from the loss of inorganic-air surface allows the organic phase to fully or partially spread over the inorganic phase, leading to core-shell or partial engulfing structures, as indicated by positive or negative $S_2$ values, respectively.

Based on Equation (2), $S_1$ and $S_2$ are defined as follows:

$$S_1 = \sigma_{23} - (\sigma_{12} + \sigma_{13}) \qquad (3)$$

$$S_2 = \sigma_{13} - (\sigma_{12} + \sigma_{23}) \qquad (4)$$

Here, $\sigma_{13}$ and $\sigma_{23}$ represent the surface tensions of the aqueous inorganic and organic-rich phase, respectively, while $\sigma_{12}$ denotes the interfacial tension at the organic-inorganic interface. Despite advances in surface tension measurement techniques, such as electrodynamic balance (Gen et al., 2023), optical tweezers (Bzdek et al., 2016; Bzdek et al., 2020) for airborne particles, and atomic force microscopy (Kaluarachchi et al., 2021) for deposited particles, determining interfacial tension between phase-separated domains remains challenging. Using SDS/AS solutions with a fixed AS concentration of 2.71 mol $L^{-1}$ as a proxy for the aqueous inorganic phase, we approximated $\sigma_{13}$ values following LLPS for systems with varying initial SDS concentrations (Figure 3c). Although more inorganic salt and SDS likely partitioned into the inorganic-rich phase during LLPS, affecting surface tension positively and negatively, respectively, accurately charactering SDS levels and simulating such high solute concentrations in macroscopic solutions presents challenges. Note that for the 1:1:0.05 and 1:1:0.5 systems, $\sigma_{13}$ is approximately equal to that of the 1:1:0.01 system due to unattainably high SDS concentrations in macroscopic solutions. While the high AS concentration (2.71 mol $L^{-1}$) significantly raises surface tension compared to SDS solutions mixed with 1.0 mol $L^{-1}$ AS, increasing SDS concentration effectively lowers the surface tension of the aqueous inorganic phase. For the estimation of organic-air surface tension, we first ignored the minor surface tension reduction caused by the enrichment of 1,2,6-hexanetriol in the organic phase during LLPS, as evidenced by Figure 3b. Given the limitations in quantifying SDS content in the organic phase, we considered two limiting cases. In the first case, we assumed that all SDS partitioned into the inorganic phase, resulting in the maximum surface tension, equivalent to that of a pure 1,2,6-hexanetriol solution at 2.71 mol $L^{-1}$ (dotted line in Figure 3c). In the second case, we assumed equal partitioning of SDS between the two phases, providing an upper limit for possible surface tension reduction (green pentacles in Figure 3c). Based on this conservative approach, we infer that increasing SDS fraction likely shifts $S_2$ values from positive to negative. Meanwhile, sufficiently low $\sigma_{13}$ values may yield a positive $S_1$, as indicated by Equation (3).



For clarity, we used insoluble organics, octane and octanal, as substitutes for the soluble 1,2,6-hexanetriol to explore the evolution of $S_1$ and $S_2$ with increasing SDS concentration. Surface and interfacial tensions for aqueous AS and organic phases were measured using the hanging droplet method. As shown in Figure 3d, with increasing SDS concentration, $S_2$
shifts from positive to negative, indicating a morphological transition from core-shell to partial engulfing (Reid et al., 2011), accompanied by an increase in $S_1$, which approaches zero at high SDS levels. Note that the surface tension of pure 1,2,6-hexanetriol was measured to be 52.02 mN m$^{-1}$, higher than the 20.72 mN m$^{-1}$ for octane and 26.36 mN m$^{-1}$ for octanol. According to Equation (3), this may result in relatively higher, and possibly positive, $S_1$ values at high SDS concentrations for mixtures with 1,2,6-hexanetriol as the organic phase.

As indicated in Section 3.1, the SDS fraction in the 1:1:0.001 mixture reached the threshold concentration, coinciding with a morphological transition from core-shell to partial engulfing. In this case, we assume that the spreading coefficient $S_2$ $\approx$ 0. Using the measured surface tensions of the simulated SDS/AS solution ($\sigma_{13}$ = 68.49 mN m$^{-1}$) and the SDS/1,2,6-hexanetriol solution ($\sigma_{23}$ = 48.49 mN m$^{-1}$) (see Figure 3c), the organic-inorganic interfacial tension ($\sigma_{12}$) is approximated to be 20 mN m$^{-1}$. Notably, $\sigma_{12}$ gradually decreased as the SDS fraction increased; hence, this conservative estimation will
maximize the parameter space for the partial engulfing morphology. Based on this, we compare the difference in liquid-air surface tensions of biphasic systems ($\sigma_{13}$ - $\sigma_{23}$ or $\sigma_{23}$ - $\sigma_{13}$) with the organic-inorganic interfacial tension ($\sigma_{12}$) to map out possible particle morphologies (Figure 4a). Furthermore, two limiting cases for the organic-air surface tensions, as mentioned above, are considered. According to surface tension measurements from SDS/AS (inorganic-phase proxy) and SDS/1,2,6-hexanetriol (organic-phase proxy) solutions (shown in Figure 3c), the corresponding blue spheres indicate the
morphology evolution from core-shell to partial engulfing across the 1:1:0.0005 to 1:1:0.5 mixtures. However, if the surface tension of the organic-rich phase is assumed to be equal to that of a pure 1,2,6-hexanetriol solution at 2.71 mol L$^{-1}$, the green spheres suggest a final inverse core-shell morphology in the 1:1:0.5 mixture, which better aligns with microscopic observations. This scenario supports the preferential partitioning of SDS molecules into the aqueous inorganic phase rather than the organic-rich phase.

Figure 4b illustrates the mixing state evolution of phase-separated 1,2,6-hexanetriol/AS particles across varying SDS concentrations. At very low SDS concentration (below 0.05% molar fraction), the equilibrium particle morphology exhibits a typical core-shell structure with $S_2$ > 0 and $S_1$ < 0. As SDS concentration increases, the decrease in $\sigma_{13}$ surpasses the reduction in the sum of $\sigma_{12}$ and $\sigma_{23}$, driving $S_2$ to negative values while $S_1$ remains negative. In this case, the aqueous surface of the inorganic phase becomes more stable, preventing full spreading of the organic phase and leading to partial organic-
phase engulfing (Reid et al., 2011). With further increases in SDS fraction, the structure shifts to partial inorganic-phase engulfing. At very high SDS levels (above 1:1:0.5 system), the sum of $\sigma_{13}$ and $\sigma_{12}$ falls below $\sigma_{23}$, resulting in positive $S_1$ with a negative $S_2$. This scenario indicates that the energy recovered from reducing the organic phase surface area is sufficient to expand both the inorganic phase surface and the organic-inorganic interface, leading to an inverse core-shell morphology. While previous studies have proposed surface/interfacial tension-driven mechanisms for mixing states in
systems comprising insoluble organics and aqueous inorganics (Reid et al., 2011; Kwamena et al., 2010), the observed




partial inorganic-phase engulfing and inverse core-shell morphologies after LLPS for atmospheric internally mixed particles are reported here for the first time.

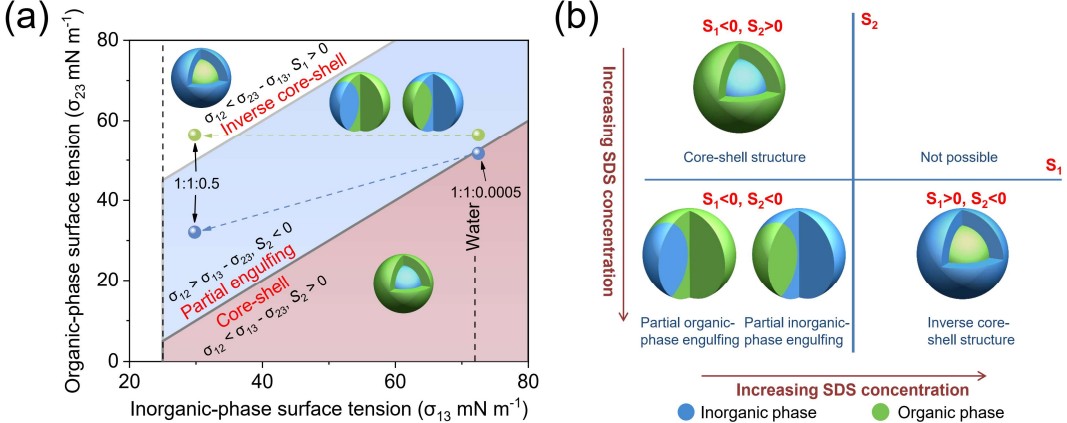

**Figure 4.** (a) Possible morphologies of biphasic organic-inorganic mixed particles, determined by the evolution of surface

tensions of the two separated phases, assuming a constant organic-inorganic interfacial tension ($\sigma_{12}$) of 20 mN m$^{-1}$. The spheres connected by dotted arrows represent the two limiting cases of organic-air surface tensions for biphasic particles with org:sulf:SDS ratios of 1:1:0.0005 and 1:1:0.5. The blue spheres correspond to measurements from simulated SDS/AS and SDS/1,2,6-hexanetriol solutions, as shown in Figure 3c, while the green spheres represent cases where the surface tension of the organic-rich phase is assumed to be equal to that of a pure 1,2,6-hexanetriol solution at 2.71 mol L$^{-1}$. (b)

Quadrant plot illustrating how spreading coefficients dominate the mixing states of organic-inorganic mixed aerosols undergoing LLPS with increasing SDS concentration.



### 3.4 Dependence of particle mixing states on surfactant types

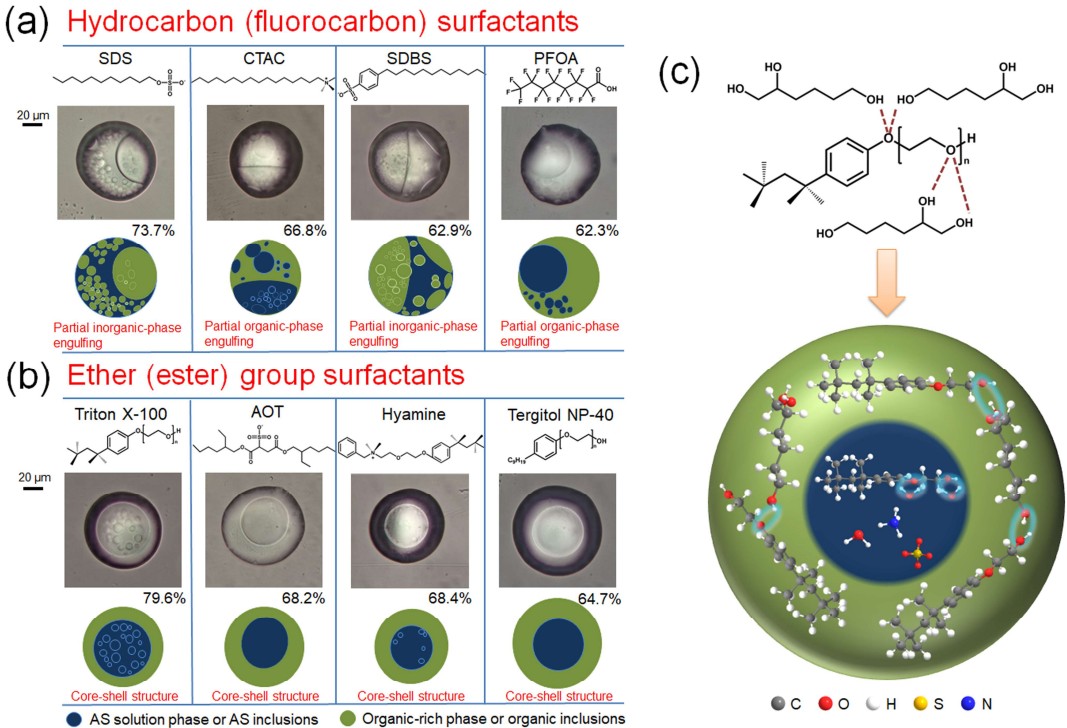

**Figure 5.** Optical images and corresponding illustrations of mixed 1,2,6-hexanetriol/AS/surfactant particles with a molar ratio of 1:1:0.1 after LLPS, incorporating different surfactant types: (a) hydrocarbon or fluorocarbon surfactants and (b) ether or ester group surfactants. (c) Schematic diagram illustrating the partitioning of ether or ester group surfactants into the organic-rich phase, demonstrated using Triton X-100, driven by hydrogen bonding with 1,2,6-hexanetriol molecules.

To validate the applicability of surfactant-mediated effects on LLPS, we examined the influence of surfactants beyond anionic SDS. Figure S12 illustrates the phase transition behaviors of particles with an OIR of 1:1 in the presence of the cationic surfactant CTAC. At a very low CTAC concentration (org:sulf:CTAC = 1:1:0.00001), partial engulfing occurs in phase-separated particles (Figure S12a). With a 500-fold increase in CTAC concentration (1:1:0.005), the morphology shifts to an inverse core-shell structure (Figure S12b). Notably, these concentrations are significantly lower than those required for SDS-induced transitions. As expected, CTAC more effectively reduces the surface tension of the AS aqueous phase than SDS at low concentrations (refer to Figure S13), leading to negative $S_2$ or positive $S_1$ values at lower thresholds. Interestingly, at higher CTAC concentrations, the particles revert to a core-shell morphology (Figure S12c). Specifically, the 1:1:0.1



mixture exhibits partial organic-phase engulfing, while the 1:1:0.5 mixture returns to a core-shell structure after LLPS. This behavior likely arises from the intricate decreasing trends of surface and interfacial tensions between organic and inorganic phases. A plausible explanation is that, at high concentrations, excess CTAC partitioned into the organic-rich phase due to

complex chemical thermodynamics and interactions with solute and water (Choczynski et al., 2024), such as its relatively low water solubility (Kim et al., 2020), thereby substantially lowering the surface tension at the organic-air interface. Combined with the inherently higher minimum surface tension of CTAC/AS aqueous phases compared to SDS/AS systems (Figure S13), this may cause $\sigma_{23}$ to be lower than $\sigma_{13}$, ultimately resulting in positive $S_2$ values.

We further investigated the effects of the nonionic surfactant Triton X-100. As shown in Figure S14, core-shell

morphology persists without partial engulfing, even in a 1:1:1 mixture. In additional experiments, perfluorocaprylic acid (PFOA), a representative PFAS compound with a hydrophobic chain similar to hydrocarbon surfactants, induces partial engulfing (Figure S15). Based on these observations, Figure 5 summarizes the final mixing states after LLPS for OIR = 1:1 particles mediated by eight surfactants. Hydrocarbon and fluorocarbon surfactants, including SDS, CTAC, SDBS, and PFOA, promote partial engulfing structures (Figure 5a). In contrast, ether or ester group surfactants like Triton X-100, AOT,

Hyamine, and Tergitol NP-40, which contain oxygen atoms in the hydrophobic chains, maintain a core-shell morphology (Figure 5b). This highlights the specific potency of hydrocarbon and fluorocarbon surfactants, lacking oxygen in their hydrophobic chains, to regulate biphasic particle mixing states.

Comparing $S_1$ and $S_2$ for model systems of octane/AS and octanol/AS with the addition of SDS, CTAC, or Triton X-100 (Figure S16), we found no significant differences. Previous studies suggested that hydroxyl groups in alcohols can form

hydrogen bonds with oxygen atoms in the polyoxyethylene chain of surfactants, influencing the cloud point and hydrophile-lipophile balance (HLB) of surfactants based on chain length and hydroxyl group number (Marszall, 1977, 1978; Gu and Galera-Gómez, 1999). Long-chain or polyhydric alcohols, such as hexanol, when added as cosurfactants, replace water molecules in micelle palisade layer, decreasing micelle hydration and surfactant hydrophilicity (Alauddin et al., 2009; Kabir et al., 2002). This will transform surfactants like sucrose monododecanoate from hydrophilic to lipophilic. Additionally, a

recent study indicated that soluble saccharides at the sea surface could form hydrogen bonds with the polar headgroups of insoluble fatty acids, facilitating their sea-to-air transfer and affecting SSA formation (Xu et al., 2023). Based on this, we infer that ether or ester group surfactants may shift into the organic-rich phase through hydrogen bonding (Figure 5c), significantly reducing surface tension at the organic-air interface and maintaining positive $S_2$ values, thereby favoring core-shell morphology. Consequently, hydrocarbon and fluorocarbon surfactants predominantly influence particle mixing states,

whereas ether or ester group surfactants have minimal impact. Gong et al. recently reported that hydrocarbon surfactants, such as dodecyl benzenesulfonate found in indoor dust and sewage sludge, largely contributed to binding activities with human transthyretin (hTTR) (Gong et al., 2024), liver fatty acid binding protein 1 (FABP1), and peroxisome proliferator-activated nuclear receptor $\gamma$ (PPAR$\gamma$) (Gong et al., 2023). These findings underscore the need to reassess the environmental implications of hydrocarbon surfactants. Additionally, integrating the effects of SDS, CTAC, and Triton X-100, we found



that low surfactant concentrations inhibited LLPS and efflorescence in mixed aerosols, while high concentrations increased

SRH and ERH values, as outlined in Section S3.

## 4 Conclusions and atmospheric implications

The reduction in surface tension and surfactant surface partitioning in atmospheric aerosols significantly influences particle

size distribution, aerosol-water interactions, and reactive gas uptake, all of which are crucial for geochemical processes like

cloud droplet activation, ice nucleation, and atmospheric heterogeneous chemistry (El Haber et al., 2024; Prisle et al., 2019;

McNeill et al., 2006; Prisle, 2021; Faust and Abbatt, 2019; Hodas et al., 2016; Bramblett and Frossard, 2022; Wokosin et al.,

2022). Our study provides new insights, demonstrating that LLPS, driven by the non-ideal thermodynamic behavior of

organic-inorganic mixed aerosols, was strongly regulated by hydrocarbon and fluorocarbon surfactants. These surfactants

modified the surface and interfacial tensions between organic and inorganic phases, determining the positive or negative

spreading coefficients ($S_1$ and $S_2$) that shaped the mixing states of phase-separated particles, including core-shell, partial

engulfing, or inverse core-shell structures. Previous research indicated that particles with an organic O:C ratio below 0.4,

characterized by hydrophobic, surface-active organics, tend to form partial engulfing or core-shell morphologies (Gorkowski

et al., 2020). However, as atmospheric oxidation increases the O:C ratio beyond 0.4, particles favor a core-shell morphology.

Since the average O:C ratio of ambient organics is around 0.52 (Canagaratna et al., 2015), atmospheric particles generally

prefer a core-shell structure rather than partial engulfing (Gorkowski et al., 2020). In contrast, our study revealed, for the first

time, that hydrocarbon and fluorocarbon surfactants could promote partial engulfing or even inverse core-shell structures in

internally mixed organic-inorganic aerosols, where moderately oxidized SOA served as the organic phase. These findings

are critical for advancing high-resolution models of equilibrium morphology of realistic atmospheric particles.

Atmospheric aerosols offer highly reactive surfaces for heterogeneous reactions with gaseous species such as $NO_2$ and

$\alpha$-pinene (Gibson et al., 2006; Song and Carmichael, 2001; Finlayson-Pitts and Hemminger, 2000; Drozd et al., 2013).

Organic coatings on phase-separated aerosols can suppress gas uptake and secondary aerosol formation (Cosman et al., 2008;

Brown et al., 2006; Drozd et al., 2013), and a core-shell morphology enriches the outer organic layer, accelerating

heterogeneous reactions with OH, $O_3$, and $NO_3$ (Shen et al., 2022). Conversely, partial inorganic-phase engulfing or inverse

core-shell morphologies may enhance gas uptake but reduce organic oxidation rates. Local electric fields at the air-water

interface of microdroplets, driven by anion enrichment and solvent polarization coupling, can enhance surface reactivity and

promote $SO_2$ oxidation (Hao et al., 2022; Liu et al., 2024; Tian et al., 2011; Salvador et al., 2003). Considering anion and

water molecule distributions, a core-shell structure may disrupt these fields, whereas an inverse core-shell morphology may

amplify them, potentially intensifying interfacial chemical reactions. Recent studies have also highlighted the competition

between interfacial and bulk-phase $SO_2$ oxidation in homogeneous droplets (Li et al., 2024a; Chen et al., 2022); however,

reaction pathways and kinetics likely shift with the formation of core-shell or inverse core-shell morphologies.





Water uptake in core-shell particles is hindered by diffusion through the outer organic phase (Davies et al., 2013), whereas partial engulfing or inverse core-shell morphologies, in which the hygroscopic phase is exposed to the particle-vapor interface, may enhance hygroscopic growth under subsaturated conditions. For example, a 1 μm droplet with an assumed inverse core-shell morphology equilibrates with ambient vapor in ~$10^{-4}$ s, compared to ~100 s for a traditional core-

shell structure (Shiraiwa et al., 2013). If LLPS persists under water vapor supersaturation, surfactant-modified morphologies can influence cloud droplet activation (Liu et al., 2018; Ovadnevaite et al., 2017), as an increased aqueous phase surface enhances aerosol CCN activity. For instance, pimelic acid/AS particles with a partial engulfing morphology exhibited activation diameters similar to AS alone, whereas homogeneous particles showed an activation diameter between that of pimelic acid and AS (Altaf et al., 2018).

Replacing a viscous organic shell with an aqueous inorganic phase also shortens equilibration timescales for gas-particle partitioning of SVOCs due to increased diffusivity in the outer shell (Schervish and Shiraiwa, 2023; Shiraiwa et al., 2013). Moreover, the resulting organic fraction and particle composition may differ in an inverse core-shell LLPS morphology (Zuend and Seinfeld, 2012; Zuend et al., 2010). Zhang et al. (2022) reported that in phase-separated particles, when the organic coating thickness to black carbon (BC) size ratio exceeded 0.24, most BC redistributed to the organic

coatings, resulting in an ~18% overestimation of BC radiative absorption in climate models. This, along with the impact of LLPS on the scattering and absorption efficiencies of brown carbon (BrC) as an organic shell (Fard et al., 2018), can yield contrasting results in inverse core-shell structures.

In summary, atmospheric surfactants influence radiative forcing and atmospheric chemistry not only directly through surface tension depression and surfactant surface partitioning but also indirectly by regulating phase-separated particle

morphology. Given the prevalence of LLPS in atmospheric particles and the widespread presence of surfactants, partial engulfing or inverse core-shell structures may be more common than previously thought. Further research is required to understand how these novel mixing states regulate heterogeneous reaction kinetics, SVOC partitioning, and the radiative absorption and CCN activity of tropospheric aerosols. Key knowledge gaps remain, including the universality of these unexpected phase-separated morphologies under real atmospheric conditions and the influence of inorganic and organic

composition, particle size, and bulk acidity on LLPS-related surfactant effects. Crucially, spatially resolved measurements of surface and interfacial tensions among phase-separated domains are urgently needed to accurately quantify spreading coefficients and elucidate their impact on LLPS morphologies.

*Data availability.* Data are available at https://doi.org/10.5281/zenodo.15560049 (Fan et al., 2025).


*Author contributions.* QL conceived the theme and wrote the manuscript with revision advice from SM and TX. YF and JS performed the experimental observations. MZ and SM prepared the figures. QL and SM performed the model calculation. All authors contributed to the data analysis and interpretation through extensive discussions.



*Competing interests.* The authors declare that they have no conflict of interest.

*Acknowledgements.* This work was supported by the National Natural Science Foundation of China (No. 42305109), Natural Science Foundation of Zhejiang Province (No. LMS25D050001), and National College Students Innovation and Entrepreneurship Training Program (No. 202411488033).

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
