# Peer review of "Surfactants regulate the mixing state of organic-inorganic mixed aerosols undergoing liquid-liquid phase separation"

_EGUsphere, 2025_

## Referee Comment (RC1)

Reviewers Comments:

The authors investigate the phase transition behaviors and resulting mixing state of phase-separated aerosol consisting of 1,2,6-hexanetriol, ammonium sulfate, and surfactants. In order to study on surfactant concentration effect on phase separation of mixture particles, phase separation including core-shell structure, partial organic-phase engulfing, and inorganic-phase engulfing is observed for particles consisting of 1,2,6-hexanetriol and ammonium sulfate by adding surfactant from very low to high concentration. it is found at very low concentrations of hydrocarbon surfactants, the organic phase partially engulfed the aqueous inorganic phase, but the inorganic phase partially or fully spread over the organic domain at higher surfactant levels. These distinctive mixing states may substantially alter the physicochemical properties, and optical properties of organic-inorganic aerosols under real atmospheric conditions, which help to bridge a critical knowledge gap regarding the role of surface tension evolution in the equilibrium particle morphology of internally mixed atmospheric particles and its potential impact on aerosol-chemistry-climate interactions. The topic is interesting and emphasis of surfactant concentration effect on phase separation in the mixed particles. However, there are lots of description on investigating LLPS, driven by the non-ideal thermodynamic behavior of organic-inorganic mixed aerosols, was strongly regulated by hydrocarbon and fluorocarbon surfactants. These surfactants modified the surface and interfacial tensions between organic and inorganic phases. There is a lack of phase separation mechanism for mixed particles consisting of 1,2,6-hexanetriol and ammonium sulfate by adding surfactant from very low to high concentration. In addition, many studies shows that ratio of inorganic: organic results in phase separation. The comparison is typical supposed to further explore this mechanism for inorganic-organic particles.

Specific comments are listed below:

Page 2 line 29-30: "These mixed organic-inorganic aerosols can undergo phase transitions, such as liquid-liquid phase separation (LLPS), efflorescence during dehumidification, or deliquescence upon humidification." Please distinguish the efflorescence and dehumidification, uniform the "efflorescence" or "dehumidification"

Page 2 line 30-31: "The resulting mixing states, such as liquid-liquid or liquid-solid equilibria," please rewrite them.

Page 2 line 46: "In addition, several other factors such as the organic fraction (Ciobanu et al., 2009; Bertram et al., 2011; Song et al., 2012a), inorganic species (You et al., 2013), drying rate (Altaf and Freedman, 2017), particle size (Ohno et al.,2023; Freedman, 2020; Kucinski et al., 2019), and bulk viscosity (Ye et al., 2025), can also influence LLPS occurrence." Please detail these effects on the LLPS occurrence.

Page 2 line 49: "However, limited studies have focused on the mixing states of biphasic organic-inorganic particles." to data, there are many studies on mixing states of biphasic organic-inorganic particles, please rewrite them.

Page 2-3 line 86-96: it seems that there is no strong motivation. Please rewrite them.

Page 11 Figure 3c, Evolution of SDS concentrations with AS or 1,2,6-hexanetriol in the mixed particles. actually, I am confused about this figure. Would be possible for mixture particles consisting of AS, 1,2,6-hexanetriol, and SDS to directly measure its surface tension?

Page 18 line 385: "high concentrations" to "high concentration" please check grammar, sentence structure, and so on.

---

## Author Comment (AC1)

**Response to Reviewers:**

- 2 Thanks for the reviewer's comments on our manuscript entitled "Surfactants regulate the mixing
- 3 state of organic-inorganic mixed aerosols undergoing liquid-liquid phase separation". The
- 4 reviewers' comments are helpful for improving the quality of our work. The responses to the
- 5 comments and the revisions in manuscript are given point-to-point below.

6

7

1

**Reviewer #1:**

- 8 1. There is a lack of phase separation mechanism for mixed particles consisting of
- 9 1,2,6-hexanetriol and ammonium sulfate by adding surfactant from very low to high
- 10 concentration. In addition, many studies shows that ratio of inorganic: organic results in phase
- separation. The comparison is typical supposed to further explore this mechanism for
- inorganic-organic particles.
- Author reply: Thank you for the reviewer's constructive suggestion. Ciobanu et al. (2009)
- identified three distinct mechanisms of LLPS in PEG-400/AS/H2O particles depending on the
- organic-to-inorganic mole ratio (OIR), i.e., nucleation and growth (OIR = 8:1 to 2:1), spinodal
- decomposition (OIR = 1.5:1 to 1:1.5) and growth of a second phase from the particle surface
- 17 (OIR = 1:2 to 1:8). Spinodal decomposition occurs in a barrier-free manner, in contrast to
- nucleation and growth, which requires overcoming an energy barrier (Shelby, 1997; Papon et
- al., 1999). In the nucleation-and-growth mechanism, subcritical nuclei form randomly within
- 20 the liquid medium and begin to grow continuously once the critical size is reached (Ciobanu
- et al., 2009). Similarly, Song et al. (2012) reported that LLPS in C7 dicarboxylic
- 22 acids/AS/H2O particles occurred via nucleation and growth, spinodal decomposition and
- 23 growth of a second phase from the particle surface when sulfate dry mass fractions are < 0.30,
- 24 0.30 to 0.60 and 0.6 to 1.0, respectively. In our previous study, the LLPS mechanisms for
- 25 1,2,6-hexanetriol/AS mixed particles with OIR values of 1:4, 1:1, and 4:1 were identified as
- growth of a second phase from the particle surface, spinodal decomposition, and nucleation
- and growth, respectively (Ma et al., 2021). However, in the present study, when the surfactant
- SDS was introduced into the 1,2,6-hexanetriol/AS mixed system, spinodal decomposition
- 29 became the only observed LLPS mechanism across all OIR values. This behavior may be

- attributed to the reduced surface energy of dispersed inorganic inclusions in the presence of
- 31 surfactants, facilitating the barrier-free occurrence of LLPS.
- 32 Lines 167-173: Previous studies have identified three distinct LLPS mechanisms in
- 33 organic-inorganic mixed particles depending on the OIR, i.e., nucleation and growth (higher
- OIR), spinodal decomposition (moderate OIR), and growth of a second phase from the
- particle surface (lower OIR) (Ma et al., 2021; Ciobanu et al., 2009; Song et al., 2012a). In the
- present study, the LLPS mechanisms for OIR = 1:4 particles (growth of a second phase from
- 37 the particle surface) and OIR = 4:1 particles (nucleation and growth) shift toward spinodal
- decomposition in the presence of SDS (Figures S3-S5), potentially owing to the reduced
- 39 surface energy of dispersed inorganic inclusions, which facilitates the barrier-free occurrence
- 40 of LLPS (Ma et al., 2021; Ciobanu et al., 2009).
- 41 2. Page 2 line 29-30: "These mixed organic-inorganic aerosols can undergo phase transitions,
- such as liquid-liquid phase separation (LLPS), efflorescence during dehumidification, or
- deliquescence upon humidification." Please distinguish the efflorescence and
- dehumidification, uniform the "efflorescence" or "dehumidification".
- 45 **Author reply:** Thank you for the reviewer's constructive suggestion. We have adopted
- reviewer's advice and revised our manuscript accordingly.
- 47 Lines 29-30: These mixed organic-inorganic aerosols can undergo phase transitions, such as
- 48 liquid-liquid phase separation (LLPS), efflorescence during decreasing relative humidity (RH),
- 49 or deliquescence during increasing RH.
- 3. Page 2 line 30-31: "The resulting mixing states, such as liquid-liquid or liquid-solid equilibria,"
- 51 please rewrite them.
- 52 **Author reply:** Thank you for the reviewer's constructive suggestion. We have adopted
- reviewer's advice and revised our manuscript accordingly.
- Lines 30-35: The resulting particle morphologies, i.e., homogeneous (aqueous or solid) and
- 55 phase-separated states, significantly influence aerosol surface properties, water uptake kinetics
- (Davies et al., 2013; Mikhailov et al., 2021), gas-particle partitioning of semi-volatile organic
- 57 compounds (SVOCs) (Ohno et al., 2022; Schervish and Shiraiwa, 2023; Shiraiwa et al., 2013;
- Zuend and Seinfeld, 2012; Choczynski et al., 2024), heterogeneous chemical reactions

- (Cosman et al., 2008; Shen et al., 2022; Drozd et al., 2013), and solar radiative forcing (Yu et
- al., 2005; Martin et al., 2004; Fard et al., 2018; Zhang et al., 2022).
- 4. Page 2 line 46: "In addition, several other factors such as the organic fraction (Ciobanu et al.,
- 62 2009; Bertram et al., 2011; Song et al., 2012a), inorganic species (You et al., 2013), drying
- rate (Altaf and Freedman, 2017), particle size (Ohno et al., 2023; Freedman, 2020; Kucinski et
- 64 al., 2019), and bulk viscosity (Ye et al., 2025), can also influence LLPS occurrence." Please
- detail these effects on the LLPS occurrence.
- Author reply: Thank you for the reviewer's constructive suggestion. We have adopted
- 67 reviewer's advice and revised our manuscript accordingly.
- Lines 48-54: For instance, in specific organic/sulfate systems, lower organic-to-sulfate mass
- ratios facilitate LLPS, whereas higher ratios suppress it (Bertram et al., 2011). You et al. (2013)
- reported that SRH values of organic-inorganic mixed particles are consistent with the
- 71 salting-out efficiencies of inorganic salts. The particle morphology of submicron
- organic/sulfate particles also depends on particle size and drying rate, with smaller particles
- 73 and faster drying rates hindering LLPS (Altaf and Freedman, 2017; Kucinski et al., 2019).
- 74 Recently, our study demonstrated that high bulk viscosity in amorphous or glassy atmospheric
- aerosols can inhibit or even completely prevent LLPS (Ye et al., 2025).
- 76 5. Page 2 line 49: "However, limited studies have focused on the mixing states of biphasic
- organic-inorganic particles." to data, there are many studies on mixing states of biphasic
- organic-inorganic particles, please rewrite them.
- 79 **Author reply:** Thank you for the reviewer's constructive suggestion. We have adopted
- 80 reviewer's advice and revised our manuscript accordingly.
- 81 Line 55: To our knowledge, numerous studies have focused on the mixing state of biphasic
- 82 organic-inorganic particles.
- 83 6. Page 2-3 line 86-96: it seems that there is no strong motivation. Please rewrite them.
- Author reply: Thanks for the reviewer's suggestion. We have adopted reviewer's advice and
- revised our manuscript accordingly.
- Lines 91-106: Therefore, comprehensive laboratory studies are urgently needed to clarify
- how surfactants influence LLPS processes and the equilibrium morphology of internally

mixed organic-inorganic particles, which may partially account for the uncertainties in the measurements of aerosol chemical reactivity as well as their direct and indirect radiative effects. The formation of SOA primarily results from the chemical oxidation of primary organic compounds and the gas-to-particle partitioning of SVOCs. Given that the average O:C ratio of SOA is around 0.52, 1,2,6-hexanetriol (O:C = 0.5) was selected to represent moderately oxidized SOA and serve as a chemical mimic of typical atmospheric SVOCs (Canagaratna et al., 2015; Lv et al., 2019). Ammonium sulfate (AS), one of the most abundant inorganic components in tropospheric aerosols, was combined with 1,2,6-hexanetriol in our previous study to investigate the dynamic process and mechanism of LLPS (Ma et al., 2021). To examine the role of surfactants in determining LLPS behaviors of atmospheric organic-inorganic particles, SDS was selected as a surrogate for atmospheric soluble surfactants because of its structural similarity to naturally occurring long-chain monocarboxylic acids, such as lauric acid (C12) (Mayol-Bracero et al., 2001; Gill et al., 1983), and its well-established surface properties comparable to those of natural surfactants (Gérard et al., 2016; Petters and Petters, 2016). In this study, we explored the phase transition behaviors and LLPS morphologies of 1,2,6-hexanetriol/AS mixed particles in the presence of SDS using microscopic imaging and confocal Raman spectroscopy. Furthermore, we systematically analyzed the influencing factors, such as the organic-to-inorganic molar ratio (OIR) and surfactant type, as well as the underlying mechanisms, including the role of spreading coefficients. 7. Page 11 Figure 3c, Evolution of SDS concentrations with AS or 1,2,6-hexanetriol in the mixed

88

89

90

91

92

93

94

95

96

97

98

99

100

101

102

103

104

105

106

107

108

109

110

111

112

113

114

115

116

particles. actually, I am confused about this figure. Would be possible for mixture particles consisting of AS, 1,2,6-hexanetriol, and SDS to directly measure its surface tension?

Author reply: Thank you for the reviewer's constructive suggestion. The surface tension of entire droplets consisting of AS, 1,2,6-hexanetriol, and SDS can indeed be directly measured using experimental techniques such as atomic force microscopy. However, in this study, our focus was to determine the surface tension of the phase-separated inorganic and organic domains rather than that of the whole mixed droplets. To the best of our knowledge, there are

currently no suitable experimental techniques capable of spatially resolving surface or

- interfacial tension between biphasic systems exhibiting core-shell or partial engulfing
- morphologies. Therefore, we prepared simulated bulk solutions based on the predicted solute
- 119 concentrations of deposited droplets to investigate the evolution of surface tension in
- separated organic and inorganic phases as a function of increasing SDS fraction.
- 8. Page 18 line 385: "high concentrations" to "high concentration" please check grammar,
- sentence structure, and so on.
- Author reply: Thank you for the reviewer's constructive suggestion. We have adopted
- reviewer's advice and revised our manuscript accordingly.

125

- 126 Reference:
- 127 Ciobanu, V. G., Marcolli, C., Krieger, U. K., Weers, U., and Peter, T.: Liquid-liquid phase separation in
- mixed organic/inorganic aerosol particles, J. Phys. Chem. A, 113, 10966-10978, 2009.
- 129 Ma, S., Chen, Z., Pang, S., and Zhang, Y.: Observations on hygroscopic growth and phase transitions of
- mixed 1, 2, 6-hexanetriol/(NH4)2SO4 particles: investigation of the liquid-liquid phase separation
- 131 (LLPS) dynamic process and mechanism and secondary LLPS during the dehumidification, Atmos.
- 132 Chem. Phys., 21, 9705-9717.
- Papon, P., Leblond, J., and Meijer, P. H. E.: The physics of phase transitions: Concepts and applications,
- 134 Springer, 1999.
- Shelby, J. E.: Introduction to glass science and technology, The Royal Society of Chemistry, Cambridge,
- 136 U.K., 1997.
- Song, M., Marcolli, C., Krieger, U. K., Zuend, A., and Peter, T.: Liquid-liquid phase separation and
- morphology of internally mixed dicarboxylic acids/ammonium sulfate/water particles, Atmos. Chem.
- 139 Phys., 12, 2691-2712.

140

---

## Author Comment (AC2)

**Response to Reviewers:**

- 2 Thanks for the reviewer's comments on our manuscript entitled "Surfactants regulate the mixing
- 3 state of organic-inorganic mixed aerosols undergoing liquid-liquid phase separation". The
- 4 reviewers' comments are helpful for improving the quality of our work. The responses to the
- 5 comments and the revisions in manuscript are given point-to-point below.

6

7

1

**Reviewer #2:**

- 8 Figure 3: It would be worthwhile to discuss the pure surface tension of 1,2,6-hexanetriol and how
- 9 it compares to lower-surface-tension organics or typical secondary organic aerosols. This
- discussion could be added near Figure 3 or incorporated into the main discussion section. My
- 11 reasoning is that a lower-surface-tension organic compound would likely remain as the
- 12 particle-engulfing phase for a longer period (as surfactant concentration increases) and may not
- transition to a core–shell morphology as readily.
- 14 **Author reply:** Thank you for the reviewer's constructive suggestion. The surface tension of pure
- 15 1,2,6-hexanetriol was measured to be 52.02 mN m-1. For aqueous 1,2,6-hexanetriol solutions, the
- surface tension decreased from 67.43 to 56.37 mN m-1 as the concentration increased from 0.17 to
- 17 2.71 mol L-1. These values are consistent with those of typical secondary organic matter in
- 18 tropospheric aerosols. For instance, McNeill's group found that hemiacetal oligomers and aldol
- 19 condensation productions formed from aqueous reactions of formaldehyde, acetaldehyde, glyoxal,
- and methylglyoxal reduced the surface tensions of pure water and AS solutions to a minimum
- value of approximately 45 dyn cm-1 (Li et al., 2011; Schwier et al., 2010). Hritz et al. (2016)
- 22 measured the surface tension of ozone-oxidized α-pinene particles using atomic force microscopy
- and reported values of 27.5 and 44.4 dyn cm-1 at < 10% and 67% RH, respectively. Similarly,
- 24 Upshur et al. (2014) reported that aqueous solutions of isoprene oxidation products exhibited the
- surface tension values around 60 mN m-1 at an organic concentration of 30 mM.
- Given that the average O:C ratio of atmospheric SOA is around 0.52, 1,2,6-hexanetriol with
- 27 an O:C ratio of 0.5 was selected to represent moderately oxidized SOA (Canagaratna et al., 2015;
- 28 Shen et al., 2022). In contrast, primary organic aerosols with lower O:C ratios and lower surface
- 29 tensions, such as decane, octanol, and oleic acid, tend to form core-shell or partial engulfing,

30 rather than inverse core-shell morphologies when mixed with aqueous inorganic phases in the 31 presence of surfactants (Reid et al., 2011). In such systems, the organic-phase surface tension ( $\sigma_{23}$ ) 32 is low, while the organic-inorganic interfacial tension ( $\sigma_{12}$ ) remains relatively high (Gorkowski et al., 2020), yielding negative spreading coefficients  $S_1$  according to Equation (3) in the manuscript. 33 34 Consequently, inverse core-shell structures are unlikely to occur, even at high surfactant 35 concentrations. Lines 313-322: Note that the surface activity of 1,2,6-hexanetriol is comparable to that of 36 37 secondary organic matter in the real atmosphere (Li et al., 2011; Schwier et al., 2010; Hritz et al., 2016; Upshur et al., 2014). For instance, McNeill's group reported that hemiacetal oligomers and 38 39 aldol condensation productions formed from aqueous reactions of formaldehyde, acetaldehyde, glyoxal, and methylglyoxal reduced the surface tensions of pure water and AS solutions to 40 41 approximately 45 mN m-1 (Li et al., 2011; Schwier et al., 2010). Considering that 1,2,6-hexanetriol represents moderately oxidized SOA, primary organic aerosols (e.g., octane and octanal) with 42 43 lower O:C ratios and lower surface tensions tend to form core-shell or partial engulfing morphologies, rather than inverse core-shell structures, when mixed with aqueous inorganic 44 45 phases in the presence of surfactants (Reid et al., 2011). In such systems, the  $\sigma_{23}$  is low and  $\sigma_{12}$ remains relatively high (Gorkowski et al., 2020), resulting in negative  $S_1$  values according to 46 47 Equation (3). Consequently, inverse core-shell structures are unlikely to occur, even at high

49

48

**50 Reference:**

- 51 Canagaratna, M. R., Jimenez, J. L., Kroll, J. H., Chen, Q., Kessler, S. H., Massoli, P., Hildebrandt Ruiz,
- L., Fortner, E., Williams, L. R., Wilson, K. R., Surratt, J. D., Donahue, N. M., Jayne, J. T., and Worsnop,
- 53 D. R.: Elemental ratio measurements of organic compounds using aerosol mass spectrometry:
- 54 characterization, improved calibration, and implications, Atmos. Chem. Phys., 15, 253-272, 2015.
- 55 Gorkowski, K., Donahue, N. M., and Sullivan, R. C.: Aerosol optical tweezers constrain the
- morphology evolution of liquid-liquid phase-separated atmospheric particles, Chem, 6, 204-220, 2020.
- 57 Hritz, A. D., Raymond, T. M., and Dutcher, D. D.: A method for the direct measurement of surface
- 58 tension of collected atmospherically relevant aerosol particles using atomic force microscopy, Atmos.
- 59 Chem. Phys., 16, 9761-9769, 2016.

surfactant concentrations.

- 60 Li, Z., Schwier, A. N., Sareen, N., and McNeill, V. F.: Reactive processing of formaldehyde and
- 61 acetaldehyde in aqueous aerosol mimics: surface tension depression and secondary organic products,
- 62 Atmos. Chem. Phys., 11, 11617-11629, 2011.

- 63 Reid, J. P., Dennis-Smither, B. J., Kwamena, N.-O. A., Miles, R. E. H., Hanford, K. L., and Homer, C.
- 64 J.: The morphology of aerosol particles consisting of hydrophobic and hydrophilic phases:
- 65 hydrocarbons, alcohols and fatty acids as the hydrophobic component, Phys. Chem. Chem. Phys., 13,
- 66 15559-15572, 2011.

75

- 67 Schwier, A. N., Sareen, N., Mitroo, D., Shapiro, E. L., and McNeill, V. F.: Glyoxal-methylglyoxal
- cross-reactions in secondary organic aerosol formation, Environ. Sci. Technol., 44, 6174-6182, 2010.
- 69 Shen, C., Zhang, W., Choczynski, J., Davies, J. F., and Zhang, H.: Phase state and relative humidity
- 70 regulate the heterogeneous oxidation kinetics and pathways of organic-inorganic mixed aerosols,
- 71 Environ. Sci. Technol., 56, 15398-15407, 2022.
- 72 Upshur, M. A., Strick, B. F., McNeill, V. F., Thomson, R. J., and Geiger, F. M.: Climate-relevant
- 73 physical properties of molecular constituents for isoprene-derived secondary organic aerosol material,
- 74 Atmos. Chem. Phys., 14, 10731-10740, 2014.